# Social Media and Artificial Intelligence: Critical Conversations and Where Do We Go from Here?

Julia Lynn Parra [1,*] and Suparna Chatterjee [2]

1 Education and Design with Learning Technology Programs, College of Health, Education, and Social Transformation, New Mexico State University, Las Cruces, NM 88003, USA
2 School of Teacher Preparation, Administration, and Leadership, College of Health, Education, and Social Transformation, New Mexico State University, Las Cruces, NM 88003, USA; suparna@nmsu.edu
* Correspondence: juparra@nmsu.edu

**Abstract:** Prior to and during the pandemic, social media platforms such as Twitter and Facebook emerged as dynamic online spaces for diverse communities facilitating engagement and learning. The authors of this article have explored the use of social media with a focus on Twitter for engagement and student-centered design of online courses in higher education. As with all technology, social media is also riddled with complex issues and unfortunately, is increasingly considered unsafe. Students have often been hesitant in their use of social media, especially for coursework and unfortunately, this hesitation has only worsened. Considering this and recent developments, social media has become a questionable tool for use in education, yet remains integral to the lives of many, both personally and professionally. The emergence and popularity of generative artificial intelligence (GenAI) tools such as ChatGPT, Lensa AI, and Canva Magic Write present new challenges and opportunities and cannot be avoided by the educational communities. Is there hope for social media and AI tools during these uncertain times? Through the combination of a current literature review and qualitative collaborative autoethnographic research, the authors take a step back and engage in critical conversations about what we have learned from our uses of social media for engagement and learning in our online courses, with a focus on (1) the intentional uses of social media, (2) the challenges and concerning issues of social media tools, and (3) exploring the implications of artificial intelligence. Centering on the theme of "hope," the authors navigate these educational and technological landscapes and answer the question *"where do we go from here?"* The authors are faculty at a southwest border university teaching preservice and in-service teachers alongside those who want to learn more about education and design with learning technologies. Their voices represent faculty, teachers, and students who are engaging with and immediately impacted by the challenges and opportunities of rapidly advancing technologies.

**Keywords:** online education; critical conversations; social media; Twitter; artificial intelligence; ChatGPT





## 1. Introduction

In the early 2020s, a series of pivotal events, for better or worse, have significantly reshaped the educational technology landscape. These included the global COVID-19 pandemic and subsequent lockdowns, which fundamentally impacted how education is delivered and experienced, the upheaval in the social media landscape, and the release of generative artificial intelligence (GenAI) tools like Chat Generative Pre-trained Transformer (ChatGPT). There are many critical questions to ask and conversations to be had about uses of the Internet, everything on the spectrum from the concerns and problems of mental, physical, and digital health/safety which have been increasingly in the spotlight, to the implications of empowerment and engagement for teaching and learning. In higher education, it is too soon to fully understand the full impact of the pandemic; however, things seem to have settled a bit in a simultaneously concerning yet hopeful aftermath. It

is a concerning time due to disruptions, failures and dehumanization in education due to the pandemic [1,2], as well as increased uncertainty and instability evidenced in existing and new technologies [3]; yet, it is hopeful because we are humans, and as we can, we will strive to take the next best steps for our students and for each other.

In this article, we explore our journey as educators and collaborative autoethnographers under a guiding pedagogy of love, care, and hope [4–7]. We will share our ongoing learning experience with each other and with our students. In facing new challenges and in the evolving landscape of social media and technology, we have adapted our teaching methods over time, shifting from a tool-centric approach to one that is student-centered and humanizes online learning. This shift is exemplified in our transition from viewing social media tools such as Twitter as technological tools and teaching from the how-to tutorial approach to recognizing their potential and affordances for empowerment and increasing engagement, such as design and teaching to align with the Community of Inquiry (COI) framework for both formal and informal learning [8].

With the juxtaposition of challenges and issues in using social media for learning and the rapidly rising popularity of GenAI tools like ChatGPT, we recognize the imperative to cultivate critical conversations in our teaching practices. As educators, and in particular educators in the fields of learning design and STEM, we have a responsibility to participate in guiding our learning communities through these new technological landscapes thoughtfully. We posit that answering the question "*where do we go from here?*" is absolutely a hopeful endeavor and prioritizes humanizing educational experiences, in this case, online educational experiences.

## 2. Methods and Context

We, the two authors of this article, are drawn to writing together due to our aligned interests in creating optimal learning experiences for our students, especially in online learning environments. We teach primarily online courses for educational technology and STEM programs in higher education. We teach both undergraduate (primarily preservice teachers) and graduate students. Our graduate students include teachers, faculty, and others with goals to become teachers or instructional designers. We meet consistently mostly via web conferencing through Zoom but also in person as we can. We use our archived online courses to engage in critical conversations, wherein we share our experiences and analyze what is and is not working from individual perspectives as well as the shared comparative experience. We review the current literature and participate in our own networks including LinkedIn, Facebook, X (formerly known as Twitter), and TikTok, and share with each other what we are encountering and learning. We started recording our conversations in the web conferencing tool Zoom, as this creates transcripts that serve as a form of reference and data collection.

In our work with additional colleagues, we identified that the best way to describe our research work is from the qualitative social science research perspective and the use of collaborative autoethnography (CAE) methods [9]. Collaborative research performed in groups and duos is quite common, but this model is unique, as rather than researching externally, "collaborative autoethnographers turn their interrogative tools on themselves, generating and utilizing their autobiographical data to understand social phenomena" [9] (p. 37). CAE "preserves the unique strengths of self-reflexivity associated with autobiography, cultural interpretation associated with ethnography, and multi-subjectivity associated with collaboration" [9] (p. 17). This approach "challenges the hegemony of objectivity or the artificial distancing of self from one's research subjects" [9] (p. 18), and the benefits of CAE include "power sharing, learning from one another, and efficiency in engaging in qualitative date collaboration" (p. 12).

When it is the two of us, our CAE approach is duoethnography: "a full dialogue model between two researchers," as discussed by Chang et al. [9] (p. 50) and is also referred to in this article as "critical conversations." In addition to our continuous, asynchronous (e.g., email, Google docs, and social media messaging), and synchronous dialogue

(e.g., Zoom, phone, and in-person meetings), we agree with identified core tenets of duoethnography, including the need to recognize differences and power differentials in support of establishing a baseline as equal collaborators and "the importance of noting the situatedness of meaning" [9] (p. 51). As we develop our manuscripts, we work both concurrently and sequentially, using Zoom and Google Docs to converse and brainstorm, write, and rewrite. This is both a challenging yet highly rewarding process, with personal outcomes that are greater than the writing and critical for our own course design and students, as we navigate the protean, opaque, and unstable digital landscape [3].

For several years, part of our ongoing dialogue and research has been focused on the use of social media, especially Twitter, for increasing interactions and presences, and thus, engagement, community development, and learning in our online courses. We currently teach online using the learning management system Canvas, and the web conferencing tool Zoom. Though the use of Twitter has been one of our key research foci, we have also used Facebook, TikTok, and Instagram. However, in 2022, Twitter was sold, renamed to X, and is continuously being radically modified, making it difficult to use with students and for the first time in years, in the fall of 2023, one of the authors did not include the use of social media in her online courses. As we were discussing these issues and this article early in 2023, GenAI technologies entered the digital landscape. These tools, especially ChatGPT, created a deep stir in education and became a part of our conversations. Thus, the purpose of this study is to explore the overarching question "*where do we go from here?*" as related to the use of social media and artificial intelligence in our courses and with our students.

## 3. Conversations with Current Literature

The call for this special issue provided a timely and relevant catalyst and backdrop to engage in critical conversations and address the disruptions and developments in social media, as well as the advent of new GenAI tools and what they mean for teaching and learning in our own online courses set for higher education. We engaged with the current scholarly and grey literature as part of our critical conversations and addressed our focus areas for this article, including what we have learned about the intentional uses of social media, the challenges and concerning statuses of social media tools, and the implications of artificial intelligence.

### 3.1. Intentional Uses of Social Media

Social media was not originally designed for educational purposes, but certain features such as Web 2.0 applications and social networking have made it a useful tool in education [10]. Twitter as a microblogging tool that includes the engaging and organizational feature of hashtags, supported professional [11] and self-directed learning [12], and has an impact on the community, communication, and casual (informal) learning for students [13]. Social media creates connections for niche communities and is often described in the context of the development of personal and/or professional learning networks (PLN) [14,15], creating broader individual and collective learning opportunities. Professional educational communities have found numerous types of interaction, engagement, and empowerment [16], and consider the criticality of Black Twitter [17], wherein exists "one of the largest gatherings of Black online users ever" and "serves as a potent example of Black digital expertise" (para 3); and some are asking, "What's going to happen to Black Twitter?" (para 2).

The literature on social media and education highlighted that the use of social media platforms such as wikis [18], Twitter [19,20], and/or Facebook [21] resulted in higher participation and improved learning in secondary and higher education [22]. In our own recent research, we looked at social media and Twitter use in our online courses. In one study, we identified "(1) evidence of cognitive, social, and teaching presence for students completing course activities using Twitter, that is, for their formal learning; and that (2) students developed course competencies during formal course activities using Twitter that supported cognitive and social presence beyond the course requirements,

that is, for their informal learning" [8] (p. 327). In a follow-up study, we identified the importance of engaging students with "(1) sharing of learning artifacts, (2) engaging in creative pedagogical practice, (3) the concept of fun, and (4) collaboration and teamwork", which confirmed for us "(1) the importance of student-centered design, (2) the continued use and adoption of relevant technology tools and skills, and (3) building community with the frameworks of Community of Inquiry and the modes of interaction model" [23] (p. 251). This successful use of Twitter encouraged us to continue revising our courses, focusing on student engagement as a priority during and immediately after the pandemic, when the challenges for online learning were amplified. However, in 2022, we were in dialogue about our teaching and students and started discussing the issues we were experiencing with our uses of social media.

*3.2. The Challenges and Concerning Status of Social Media*

Until recently, the perceived benefits of integrating social media into education outweighed its disadvantages [24,25] prompting many educators, like us, to adopt it as a tool for facilitating engaged learning [26]. Of course, using social media in education has had its issues. Challenges have included classroom distractions [27], a perceived loss of control over students in the classroom [28], and reduced student focus and multitasking negatively impacting learning, performance, and retention [29]. Additionally, concerns have arisen among students who are not comfortable using social media due to concerns of social media addiction and cyberbullying [30–33], issues related to security and privacy such as the public accessibility of information [34], and an inundation of misinformation and disinformation [35].

Sundaram and Radha [36] investigated the security involved in social media use among youth internet users. They found that social networks store end users' information remotely to personalize services and sell information to advertisers. These practices raise concerns about privacy and the commodification of personal information and contribute to the accumulation of "Big Data" [36]. Big data has been identified both as a priority and concern since 2014–2016 by the policy of the Obama White House Office of Science and Technology [37]. Moreover, this use of automated and algorithmic processes in social media has led to concerns about unintended bias and discrimination [38,39], which can be perpetuated through academic texts [40,41] and news outlets [42]. Bias in machine learning has been discussed by researchers [43,44], and experts argue that these technologies are not neutral; rather, they are value laden [45–47] and their design has the potential for "racialized, gendered and colonized hierarchies" [37] (pp. 2123–2124). These issues have become increasingly present in recent years, particularly with significant events such as the COVID-19 pandemic, the sale of Twitter, and the rise of GenAI tools. Author B encountered increased concerns about the use of social media from students in an undergraduate course in 2022 and author A was experiencing increasing challenges using Twitter in her classes; both authors have felt the need to step back and hold critical conversations with each other and their students. As humans and educators, it is incumbent upon us to take responsibility and be accountable for the outcomes we are all experiencing. With increasing challenges and concerns in the social media landscape, educators and educational researchers are taking a step back, and with the advent of generative AI tools that carry many of the same issues alongside new challenges, Mishra et al. note both "hand-wringing-and some celebration-about the impact these tools will have on education" [3] (p. 235).

*3.3. The Implications of Artificial Intelligence*

Artificial intelligence in education has been the subject of research for over two decades [48]. However, it was not until the past couple of years (2022–2023) that AI tools, specifically generative AI tools like ChatGPT, DALL-E, MidJourney, Bard, Bing Chat, Lensa AI, and Canva Magic Write [3,49,50], became widely accessible and started to influence online teaching practices. ChatGPT was released by OpenAI to the world in the late fall of 2022, and at the time, it was estimated to have reached "100 million monthly active

users in January 2023" [51] and was noted to be "the fastest-growing consumer application in history" (p. 1). Watters and Lemanski [52] conducted a review of the early literature on ChatGPT, with findings revealing a "predominance of negative sentiment across disciplines" and "raising concerns about employment opportunities and ethical considerations" similar to concerns of use of social media and the internet in general of "privacy, bias, transparency, and accountability", yet holding "promise for improved communication" and needing further research "to address its capabilities and limitations" (Abstract and Discussion para 2). Dai, Liu, and Lim [53] identify ChatGPT as "a student-driven innovation" (p. 1) and a "potent enabler for enhancing education quality and transforming higher education" specifically, as it and tools like it "can be leveraged to enhance learning analytic techniques, generate customized scaffoldings, facilitate idea formation, and eventually expand educational access and resources for social justice" (p. 2).

Sok and Heng [54] highlighted some time-saving educational uses of ChatGPT, including helping teachers develop learning assessments, provide virtual tutoring, draft outlines, and brainstorming. They identified concerns related to such uses of ChatGPT, especially in regards to academic integrity including biased learning assessments, inaccurate or fake information, and an overreliance on AI tools. For example, using AI for brainstorming an idea or to create an outline could interfere with students developing these skills as well as losing the practical experience of becoming successful after struggle [55]. Part of the art of teaching and learning is scaffolding student learning and balancing it with the right amount of struggle, i.e., through the zone of proximal development [56]. If artfully used, these burgeoning GenAI tools might support scaffolding and assistance to the struggling learner, creating the opportunity for learning at the early stages where a student might give up, thereby facilitating and deepening learning experiences, e.g., "get away from the high school paper and go further, to write something larger, like a thesis" [55] (para 33).

The impact of GenAI on educational practices is in its early stages, and it is ChatGPT that is generating most of the discussion. The discussions cross the spectrum from the language of opportunity, time-saving strategies and efficiencies, hopeful transformations, and the potential to revolutionize education [57–59] to the language of challenges and fears; again, mostly regarding how assessments will be impacted and long-held concerns related to cheating and plagiarism [58], but also vulnerabilities related to bias, dis- and misinformation, and cybersecurity and privacy [52,60,61]. Fullan, Azorín, and Harris [58] note that "an assessment of the real impact that this technology will have on teaching and learning for good or bad, has yet to be made," that "there is a lack of research, guidelines, and regulations specific to ethical issues raised by the application of GenAI to education," (p. 2) and that there is a tangible fear regarding "whether AI in education has been designed to supplant teachers/leaders or reduce them to a functional role, rather than to assist them to teach/lead more effectively" (p. 5).

Of note, two key publications have been especially instrumental as we engaged in dialogue and critical conversations. The first, "TPACK in the age of ChatGPT and Generative AI" [3] was a product of interinstitutional coauthoring by one of our colleagues within our department who shared it with us. In this article, Mishra et al. [3] highlighted the need to further develop "TPACK in the age of Gen AI," (p. 247) arguing for a "more expansive description of contextual knowledge (XK)" (p. 236) that accounts for the broader implications of GenAI on individuals and society. This work provided key essential descriptions and terminology, including a description of GenAI as "applications which are designed to create new content (text, images, video, music, artwork, synthetic data, etc.)" (p. 236). Additionally, they offered a set of probing questions that enriched our critical conversations. They note that these questions should have been "asked of social media over a decade ago" (p. 237) and we agree, as we step back from our own uses of social media. The first questions in their list are "What does it mean to teach in an era where GenAI becomes part of our everyday life? In a time when it will be increasingly difficult to distinguish between AI-generated and human-generated content?" (p. 237).

The second publication, "How do we respond to generative AI in education?" by Mills, Bali, and Eaton [62], proposes that open educational practices "can help educators cope and perhaps thrive in an era of rapidly evolving AI" (p. 16). It was shared with one of us on LinkedIn and begins to address the aforementioned questions by advocating for open educational practices, two of which stood out for their immediate relevance to this study: engaging with interdisciplinary and interinstitutional online communities for ideas exchange and reflection and collaborating with students. These practices are not just theoretical, as they are the very means by which these articles reached us, exemplifying the power of open educational resources. Furthermore, the practice of collaborating with students has been crucial for us in answering the question "*where do we go from here?*" In Section 4, author A provides an autoethnographic narrative reflecting and responding to this question.

## 4. Where Do We Go from Here? Narrative Reflection and Response

Dede and Lidwell [63] note that "AI is becoming increasingly proficient at calculation, computation, and prediction ("reckoning") skills" and forecast that "we will see increased demand for human judgment skills such as decision making under conditions of uncertainty, deliberation, ethics, and practical knowing." They challenge us "not merely to understand how remote learning and AI can scale present capabilities, but to also use this moment to reflect and reimagine the learning experiences of students" (p. 7). Author A engaged with the current literature and reflected on her experiences this year, 2023, with online teaching and learning in her classes and with her students.

### 4.1. If the Robots Take Over, Shame on Us!

I joined Twitter in 2006 and have been formally using Twitter in my courses since 2011 when I created an activity: the Twitter Top 5, which combined developing collaborative teamwork skills and developing personal learning networks (PLNs) alongside the exploration of Twitter. With my classes (I teach in fully online programs), we would create what we would refer to as a "community bubble of safety", and in addition to the Twitter Top 5, students shared their learning artifacts and their creations such as infographics, concept maps, comic strips, etc., and synthesized reflections based on their "aha!" discovery moments in class; they engaged in fun meme wars and the use of hashtags for interaction and networking. I think it is also important for me as an educator to enjoy the process of teaching and learning, and engaging with my students on Twitter did that for me as well. Of note, there was one semester that I tried TikTok instead of Twitter, and it just wasn't the same community building experience, as it lacked the ease of use that Twitter provided.

Upon examining our courses and reviewing the effectiveness of our social media strategies, particularly in our use of Twitter as highlighted in Section 3.1, I found it affirming to identify the successful aspects of these practices. Twitter has served as an exemplary platform for teaching social media dynamics, and only by helping others use this tool was I able to explain to others why it was continuously highly ranked and was the #1 top ranked tool for seven years (2009–2015) on the Top 100 Tools for Learning [64] list that I have been tracking since 2007. Twitter remained in the top 20 until this year, 2023, when it dropped to number 22. There was another tool that upon release, immediately entered the 2023 Top 100 Tools for Learning [64] list at number 4: ChatGPT. By spring 2023, when ChatGPT was introduced, I was having numerous conversations with colleagues and students about what was happening in social media. There were senate hearings about TikTok and the potential banning of TikTok, and as Twitter was sold and renamed X (I call it TwiX), the social media experience for student learning was quickly deteriorating.

For example, in my summer class, my posts and my students' posts would intermittently not publish, and with constant changes within the platform, its instability tipped the scales in disfavor of use, and I was literally pondering this question of "*Where do we go from here?*" In the summer of 2023, with the articles from Mishra et al. and Mills et al. in hand, I engaged my online social media in education class in a transformative process. We

collectively examined the role of social media and technology in our lives and academic endeavors. Through participatory design during our live class meetings, we co-designed new class activities and projects. These included developing personal social media and tech health plans, implementing individual pathways of learning to explore artificial intelligence, and creating team presentations on the topic of a digital bill of rights. Additionally, we discussed our developing ideas about the values and norms we should be thinking about regarding student use of artificial intelligence tools in our programs and courses, and started the Building on Class AI Values and Norms document (see Appendix A) that I now use with all of my classes.

And so, for the first time since 2011, in the fall of 2023, I chose not to incorporate social media platforms into my teaching. What I have done is keep and expand upon the many things I have learned from using social media, with a focus on humanizing online learning [65] and the use of student-centered design models. I also continued to employ participatory design [66,67] and cocreation [67,68], and prioritize critical conversations. Engaging in critical conversations can mean conversations that are important and timely in topic and/or it can mean critical in process. In either case, they require scaffolding, and I do this with the use of community-building strategies to foster a safe and trusting space. While the COI framework remains a foundational model in my approach, I also introduce my students to the concept of an innovative knowledge-building community [69] as a comprehensive framework for these critical humanizing strategies.

Moreover, the shift away from social media has allowed for the greater exploration of emerging technologies like virtual reality (VR) and AI, which have captured the interests of students, particularly when they have the autonomy to choose their topics. For example, in my first 8-week, online, fall 2023 class, where students formed two teams to design, develop, and deliver webinars, both teams chose to focus on AI in education. It is important to note, that as the learning designer, my own focus on these tools is influential on my students' choices, and students from the summer class were in this fall class. In this class, our formal interaction with AI was focused on my redesigned syllabus and an orientation webinar where we discussed class AI values and norms.

I keep an open discussion for students to ask me any questions they have, and I had shared that I was working on this article, and one student asked me to share my thoughts at the time about AI and social media. This was my response:

"Hmmm, as you probably already know, AI caught all of our attention. So, integrating that in the social media course and webinars courses and seeing everyone take off with it, was so great! I do think the VR/AR tools are going to be something to watch for though the expense is an issue as is true with a lot of digital scenarios. The collaboration between Meta and Ray Ban AR might be interesting.

By adding Twitter in the past, that one thing gave us some fun interaction and I loved it. Now I have to rethink my social media scenario as Twitter implodes. TikTok was fun too but there's something too much with TikTok that I'm also unsure about. So, I am falling back on individualization/personalization strategies that I see students learn so much from. I am referring to the 1.6–4.6 activities in this course. It might not be as fun, but I see the engagement.

So, I mostly think our theoretical and conceptual frameworks that drive our strategies for engagement are what are most important. I'm still working on this but I would say that my theoretical framework includes the Pedagogy of Love, Care, and Hope and my conceptual framework include a blended focus on Universal Design for Learning (UDL), and TPACK along with the presence and interaction models. Under those frameworks I would note powerful strategies for engagement such as gameful design, community building, collaboration, teamwork, synchronous interaction, reflection, etc. We can really do a lot with the basics of Canvas, Zoom, and Google Docs with these strategies. Also, I have worked with my students through participatory design a lot over the years to make discussions and activities more engaging" [70].

I have adopted and shared with my students the phrase "If the robots take over, shame on us! For we did not do enough to humanize education." This has helped us to create a focus in our class conversations on what we hope remains key to designing optimal learning experiences: the humans.

### 4.2. Modeling and Disclosure of Use of ChatGPT

In my classes this year, I modeled and discussed my use of ChatGPT, and as noted in Appendix A: Building on Class AI Values and Norms, we all agreed on the importance of transparency and disclosure when we use GenAI tools, as well as citing when we reference ChatGPT. Here is our disclosure of use in this article, and the quote below from ChatGPT is added to our references.

I use the ChatGPT Plus plan. As we wrote this article, I asked it for three types of help. First, sometimes I wanted ideas to reword something that I was trying to say. For example, for the first sentence of the Introduction, I wrote, "In the early 2020s, several things occurred that, for better or worse, have completely altered the educational technology landscape including" and I asked ChatGPT for a different wording, and it gave me "In the early 2020s, a series of pivotal events, for better or worse, have significantly reshaped the educational technology landscape." Second, I asked it to provide an editorial review of our literature review, and we updated it with a few of the recommendations. For example, in the Implications for Artificial Intelligence section, we started with only the two key publications, and ChatGPT recommended "Discussing the implications of AI in education in more depth, including both opportunities and challenges, would enrich the narrative" [71]. And third, we normally work in APA, and rebuilding the References section was tedious. So, I gave ChatGPT the references numbered 30–74 to help format it. It provided most of the formatting, and then we went through bolding the journal years, checking the abbreviations of some of the journal names, and adding the locations and dates of the conference proceedings publications.

## 5. Discussion

Facing the dynamic nature of educational technology, our goal as educators is to strategically shift and adapt, remove barriers, address critical needs, and foster robust support systems for our students that can advance education in ways that positively impact employment and job satisfaction. As we navigate the evolving landscape of technology in our personal and professional spheres, we remain committed to exploring innovative ways to integrate social media and AI into meaningful learning experiences, most of which, in our cases, are online. Our journey has shown us the benefits of social media in enhancing student interaction, fostering COI, and developing personal learning networks [7,8,23], thereby increasing engagement and supporting student learning.

We envision a future where our understanding of technology's transformative potential is matched by our ability to apply it effectively in communication, education, and problem solving. Central to harnessing this power is the emphasis on collaboration and community building through critical conversations. Our focus is not merely on whether to use specific tools, but on empowering learners, educators, practitioners, and researchers to utilize these tools effectively. We strive to go beyond mere interface adjustments, prioritizing practices that humanize the educational process by openly addressing issues, fostering community, and designing interactive activities and assessments. This approach will help mitigate ethical concerns, support academic integrity, and enable us to achieve our educational aspirations.

Our approach to teaching and learning with technology is balanced with a critical awareness of safety and mental health considerations. We continuously reflect, engage in research, and implement participatory course design, embodying the pedagogy of love, care, and hope. Even as we may step away from certain tools, we continue to explore new ways to engage the students utilizing our pedagogies, strategies, and tools relevant for the times. These tools might be tried and true like institutional learning management

systems, or they might be emergent like VR and AI that require our engagement to support our communities. In regard to ChatGPT and other similar tools, simply prohibiting AI use is not a solution. Instead, we encourage collaborative exploration with students to discover strategies for learning with AI, leveraging its strengths for tasks like summarizing, editing, brainstorming, and receiving feedback. One essential skill has arisen, prompt engineering, where one can achieve a vast array of tasks and engage in a productive communicative interaction with these tools. Designing courses that involve students in ways that go beyond AI's capabilities, discuss objectives of assignments which require individual learners' perspectives, and clarify academic integrity and the ways to adhere to it will create value when designing, teaching, and learning with technology.

We recognize the success of our teaching through student participation and their engagement in our courses, their participation in and cocreation of a vibrant knowledge-building community, providing tools such as Twitter and ChatGPT and resources such as OERs that are creatively utilized by students, and we revise our primarily online courses based on student feedback. Students cocreate class activities and complete the activities, and students are producers in our classes. We design classes in a way for students to be successful, learn educational theory alongside instructional design, and we provide a statement in the syllabi that grades are conceptualized as progress updates similar to gameplay, and our primary goal is to excite students about the work and make learning contextual and meaningful. The context involves students from beginning to end becoming part of the design, completing the task of creating activities, analyzing, and reflecting.

A recent example of the power of these strategies is a project in one of our classes that is focused solely on online teaching and learning. Students were put in teams of 4–5, and each team was assigned to create a microlearning online course to include a document of standard operating procedures. This was a daunting task for an 8-week course, but they were provided appropriate content, scaffolding, resources, and instructor coaching. The use of GenAI was addressed at the beginning of the course (see Appendix A). The use of GenAI was discussed as questions arose and was indicated in SOPs and courses as relevant. Each team was successful with these tasks, and at the end, proudly presented outstanding final products (SOPs and fully developed microlearning courses) in our end-of-course live class meeting in Zoom.

## 6. Vision for the Future of Online Education with Social Media and AI Technologies

As we envision the future of online education with social media and AI technologies, we recommend and advocate for a widespread digital and media literacy education, enhanced cybersecurity training, in-depth discussions on core issues, and the development of effective usage policies. Professional development for educators on integrating these tools into the classroom is crucial. We encourage teaching students to use these tools responsibly, adhering to ethical standards. Tailoring the codes of conduct to course levels is important, as the applicability of tools can vary. Emphasizing the proper citation and disclosure of AI assistance is essential. We also recommend designing assignments that require not only writing but critical thinking, thus, promoting learning even with technological assistance. Grading criteria should focus on aspects challenging for AI to replicate, such as originality, emotional depth, metacognition, and personal experiences [55]. Finally, transparent institutional policies should be established, allowing for research and experimentation.

Finally, our answer to "*where do we go from here?*" includes a vision for the future of online education, where pedagogy is deeply intertwined with practice and is focused on approaches that prioritize humans and humanizing teaching and learning. Under an overarching pedagogy of love, care, and hope, our practice is rooted in the principles of participatory design and cocreation of collaborative and individualized learning experiences, where learners are active contributors to their educational journeys. We advocate for design practices that embrace the Universal Design for Learning (UDL), which promotes multiple forms of engagement, representation, and action/expression [72], along with fostering online learning communities as modeled by the COI framework [8,73] and innovative

knowledge building communities [69]. In these online learning communities, students feel safe and empowered, and in a state of innovation and cocreation, they can and will collaborate with us to continuously help re-envision and redesign education.

While pedagogy and design are at the forefront, purposeful selection and use of technology is essential. Hope is not found within the tools, but within the humans who wield them and who make the critical educational choices. In this case, we chose to step back from social media tools that we had learned to step up engagement with, primarily Twitter and Facebook, and leveraged established technologies like our learning management system, Canvas with its integrated Canvas Studio, and the web conferencing system Zoom. Additionally, we incorporated tools for student creation like Canva, as well as explored the potential of newer technologies, such as ReadyPlayerMe in preparation for virtual reality and GenAI tools like ChatGPT to engage learners. TwiX is still there, in the background, under discussion, and maybe we will use it again in the future, or maybe not. What we will do is focus on the humans and on the relationships, and never let the robots take over! If the robots take over, shame on us! For we did not do enough to humanize education.

**Author Contributions:** Both authors contributed to the conceptualization of all phases of the work. All authors have read and agreed to the published version of the manuscript.

**Funding:** This research received no external funding.

**Institutional Review Board Statement:** The study was conducted in accordance with the Declaration of Helsinki, and approved by the Institutional Review Board of New Mexico State University (protocol code 2206001085, date approved 8/18/23)" for studies involving humans.

**Informed Consent Statement:** The authors are the participants and their consent is given.

**Data Availability Statement:** This is a collaborative autoethnography and the data is primarily the voices of the authors shared within this document. The relevant online course data is provided in the Appendix A.

**Conflicts of Interest:** The authors declare no conflicts of interest.

## Appendix A

### Building on Class AI Values and Norms

- Values: integrity, responsibility, accountability, ethics, critical thinking
- AI is useful as a tool and an affordance, like we see with the Internet itself and tools like Google, Google Scholar, etc. We are encouraged to try/explore/experiment with new concepts and technologies, and in this case, we focus on artificial intelligence (AI).
- We will keep each other informed about exciting, interesting, scary/concerning, etc. things we encounter and encourage each other in this process.
- If we don't know something, we will just ask. Not only the professor but each other (there will always be space and time for this).
- AI should be recognized as riddled with issues, inaccuracies, incompleteness, biases, etc. We must explore and identify these issues. For example, you will hear Dr. Parra say, "Chat GPT is actually a liar!" and is noted in the research as being "notorious for generating text with 'hallucinations" [74].
- We must remain vigilant and mindful of relevant issues and ethics. Note that we must use these technologies ourselves to be knowledgeable and lead the way. We will be respectful, continually investigate the relevant ethics, and work within ethical uses to the best of our ability.
- There can be a thin line when it comes to AI and plagiarism similar to the use of research journals, online resources, etc. AI should not be used to do the work for you. Use AI as a tool and do not copy/use it verbatim. Be transparent.
- Disclose when AI is used. Use APA in our program to cite your AI use.
- Cross reference any images used/provided for potential copyright issues, and as relevant, provide any relevant citations.

- Apply digital citizenship and literacy knowledge. Use critical thinking. Stop and question for all of the above.

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
