# Peer review of "Social Media and Artificial Intelligence: Critical Conversations and Where Do We Go from Here?"

_education, doi:10.3390/educsci14010068_

Round 1

Reviewer 1 Report

Comments and Suggestions for Authors

Reviews Report

Manuscript ID: education-2744743

Title:  Social Media and Artificial Intelligence: Critical Conversations and Where do we go from here?

In this manuscript, the aim is to conduct a comprehensive exploration and critical analysis of the intersection between social media and artificial intelligence (AI) in the context of higher education. The authors focus on the evolving landscape of social media platforms, with a specific emphasis on Twitter, as dynamic online spaces that have played a role in facilitating engagement and learning, particularly in the context of online courses during and prior to the pandemic. The manuscript aims to address the dual nature of social media as both a valuable tool for fostering engagement and a platform fraught with complex issues, raising concerns about its safety. The hesitancy among students to use social media, especially for coursework, is highlighted as a challenge that has intensified over time. In light of recent developments, the manuscript also explores the questionable utility of social media in education while acknowledging its continued significance in personal and professional lives. The emergence of generative artificial intelligence (GenAI) tools, such as ChatGPT, Lensa AI, and Canva Magic Write, is recognized as a factor presenting new challenges and opportunities that cannot be ignored by the educational community. Through a combination of a current literature review and qualitative collaborative autoethnographic research, the authors aim to engage in critical conversations about the intentional uses of social media, the challenges and concerning issues associated with social media tools, and the implications of artificial intelligence in the educational context. The central theme of "hope" serves as a guiding principle as the authors navigate the educational and technological landscapes, ultimately seeking to answer the overarching question: Where do we go from here? The manuscript aims to provide insights and recommendations for educators and educational communities facing the complexities and uncertainties inherent in the intersection of social media and artificial intelligence.

The manuscript is very well prepared and interesting.

The topic is highly original and relevant in the field. 

The references are appropriate and relevant.

This study has the potential to be cited.

I recommend to the Editorial Office to consider this manuscript after minor revision.

Reviewer’s Suggestion

The authors' names are missing. Please add.

Although the manuscript is very well-prepared and the topic is thoroughly discussed, the authors could consider adding more on the following issues:

-Identify and analyze the challenges associated with the use of social media in higher education.

-Investigate student hesitancy and concerns, especially in the context of coursework, and how these challenges have evolved over time.

-Assess the impact of generative artificial intelligence (GenAI) tools, such as ChatGPT, Lensa AI, and Canva Magic Write, on educational practices.

-Examine both the opportunities and challenges posed by AI tools in the context of online courses and engagement.

-Investigate the safety concerns associated with social media platforms and their implications for educational settings.

-Consider ethical considerations related to the use of social media and AI tools in higher education.

-Investigate how the theme of "hope" is applied within the context of social media, AI tools, and online education.

-Explore potential positive outcomes, solutions, and strategies for addressing challenges and concerns.

-Provide insights into the future role of social media and AI tools in education.

-Offer recommendations for educators, institutions, and policymakers on navigating the evolving landscape and maximizing the benefits while mitigating risks.

The conclusion should be provided.

Comments on the Quality of English Language

Minor changes. 

Author Response

The authors' names are missing. Please add.

  • This is completed.

Although the manuscript is very well-prepared and the topic is thoroughly discussed, the authors could consider adding more on the following issues:

-Identify and analyze the challenges associated with the use of social media in higher education.

  • This was there in 3.2.

-Investigate student hesitancy and concerns, especially in the context of coursework, and how these challenges have evolved over time.

  • This was there in 3.2.

-Assess the impact of generative artificial intelligence (GenAI) tools, such as ChatGPT, Lensa AI, and Canva Magic Write, on educational practices.

  • Two paragraphs added with a focus on the current discussions regarding ChatGPT in 3.3.

-Examine both the opportunities and challenges posed by AI tools in the context of online courses and engagement.

  • Two paragraphs added with a focus on the current discussions regarding ChatGPT in 3.3.

-Investigate the safety concerns associated with social media platforms and their implications for educational settings.

  • This was there in 3.2.

-Consider ethical considerations related to the use of social media and AI tools in higher education.

  • This was there in 3.2.and more added to 3.3.

-Investigate how the theme of "hope" is applied within the context of social media, AI tools, and online education. This was there in 3.2.and more added to 3.3.

  • Discussion section is added with the theme of hope further addressed.

-Explore potential positive outcomes, solutions, and strategies for addressing challenges and concerns.

  • Two paragraphs added with a focus on the current discussions regarding ChatGPT in 3.3. A new 5. Discussion section is added addressing our potentials.

-Provide insights into the future role of social media and AI tools in education.

  • We attempt this in sections 5 & 6.

-Offer recommendations for educators, institutions, and policymakers on navigating the evolving landscape and maximizing the benefits while mitigating risks.

  • Revision includes addition of new Discussion and prior discussion is now Conclusion with an additional recommendations paragraph added

-The conclusion should be provided.

  • Revision includes addition of new Discussion and prior discussion is now Conclusion with an additional recommendations paragraph added.

Reviewer 2 Report

Comments and Suggestions for Authors

Easy to follow, nice work!

Please see the proposed corrections/suggestions in the attached file.

Thank you.

Author Response

Thank you for your conscientious, detailed review. We addressed all of your suggestions and recommendations first. 

Reviewer 3 Report

Comments and Suggestions for Authors

- More information about the contribution of this article must be added to the abstract.

- Justifications for using the research method must be clarified.

- More details about both authors should be given.

- Both authors were optimistic about the involvement of technology, but what about students' experience?

- How did the author recognize the success of their teaching? More information about the implementation context should be added.  

Comments on the Quality of English Language

Suitable level

Author Response

  • More information about the contribution of this article must be added to the abstract.
    • Two sentences added to the end of abstract.
  • Justifications for using the research method must be clarified.
    • Two sentences added to Section 2. Methods and Context, end of second paragraph.
  • More details about both authors should be given.
    • Two sentences added to Section 2. Methods and Context, in first paragraph.
  • Both authors were optimistic about the involvement of technology, but what about students' experience?
    • As evidence of students' involvement with technology in courses we cited our other research and publication. This article primarily focuses on “our” (instructors) conversation which has a foundation of how we have used these tools for years with our students and we have supported it with students’ experiences in our publications.  In this article, students are referenced in 4.1. Students are discussed as taking part in a transformative, collaborative, and participatory process, co-designing new class activities and projects, etc. and there is a new discussion section with a paragraph added about a class/students’ experience.
  • How did the author recognize the success of their teaching? More information about the implementation context should be added.  
    • Revision includes addition of new Discussion where this is discussed.

Round 2

Reviewer 3 Report

Comments and Suggestions for Authors

N/A

Comments on the Quality of English Language

N/A